# Deep Ensembles Secretly Perform Empirical Bayes

## Abstract

Quantifying uncertainty in neural networks is a highly relevant problem which is essential to many applications. The two predominant paradigms to tackle this task are Bayesian neural networks (BNNs) and deep ensembles. Despite some similarities between these two approaches, they are frequently understood as fundamentally different. BNNs are often touted as more principled due to their reliance on the Bayesian paradigm, whereas ensembles are perceived as more *ad-hoc*; yet, deep ensembles tend to empirically outperform BNNs, with no satisfying explanation as to why this is the case. In this work we bridge this gap by showing that deep ensembles perform *exact* Bayesian averaging with a posterior obtained with an implicitly learned data-dependent prior. In other words, deep ensembles are Bayesian, or more specifically, they implement an *empirical Bayes* procedure wherein the prior is learned from the data. This perspective offers two main benefits: (*i*) it theoretically justifies deep ensembles and thus provides an explanation for their strong empirical performance; and (*ii*) inspection of the learned prior reveals it is given by a mixture of point masses – the use of such a strong prior helps elucidate observed phenomena about ensembles. Overall, our work delivers a newfound understanding of deep ensembles which is of interest in it of itself, and whose insights we hope will drive future empirical improvements.

## 1 Introduction

Deep learning has proven extremely successful at prediction (Krizhevsky et al., 2012; Szegedy et al., 2015; He et al., 2016), representation learning (Mikolov et al., 2013; Chen et al., 2020; Radford et al., 2021; Caron et al., 2021; Oquab et al., 2024), and generative modelling (Radford et al., 2018; Brown et al., 2020; Ramesh et al., 2021; Rombach et al., 2022; Saharia et al., 2022; Yu et al., 2022; OpenAI, 2023), yet it still lags behind statistical methods (Kutner et al., 2005; Rasmussen & Williams, 2005; Gelman et al., 2013) at uncertainty quantification (UQ) – for example, off-the-shelf neural network classifiers can be poorly calibrated despite being accurate (Guo et al., 2017). The two most popular approaches for UQ in deep learning are Bayesian neural networks (BNNs; Welling & Teh, 2011; Graves, 2011; Hernández-Lobato & Adams, 2015; Blundell et al., 2015; Gal & Ghahramani, 2016; Ritter et al., 2018) and ensembles (Lakshminarayanan et al., 2017). BNNs operate by specifying a prior distribution over network parameters and then following the rules of Bayesian inference to obtain a posterior distribution over these parameters. The corresponding posterior predictive distribution – i.e. the expected predictive density over this posterior – then captures all relevant uncertainty. Ensembles also average predictive densities, but they do so over models trained with different seeds. BNNs are typically thought of as the more theoretically justified of the two approaches due to their adherence to Bayesian principles, with ensembles often being considered non-Bayesian due to the fact that the corresponding average is (seemingly) not performed over a posterior distribution. Nonetheless, despite being considered more *ad-hoc* and lacking formal justification, ensembles typically outperform BNNs (Abdar et al., 2021).

Providing reliable uncertainty estimates is fundamental in many scientific and safety-critical applications such as medical diagnostics (Esteva et al., 2017), autonomous driving (Bojarski et al., 2016), medical imaging (Litjens et al., 2017; Adnan et al., 2022), and more (Psaros et al., 2023); a lack of such estimates hinders widespread adoption and deployment of neural networks in these domains. A key step towards this goal is to gain an improved understanding of UQ methods in deep learning; why they work, how they are justified,

and how they relate to each other. Such an understanding is itself of scientific interest, and insights that arise from it are likely to improve UQ in the future.

In this work, we rely on *empirical Bayes* (Robbins, 1956; Carlin & Louis, 2000), a class of Bayesian procedures where the prior is learned rather than pre-specified. More specifically, we show that deep ensembles are *equivalent* to BNNs where an arbitrarily flexible learnable prior is obtained through maximum marginal likelihood. This result is satisfying on several fronts. First, it shows that ensembles are an empirical Bayes procedure, which justifies ensembles. Second, it sheds light on how these two prominent paradigms are much more closely related to each other than previously realized. Lastly, our analysis reveals that the prior implicitly learned by ensembles is particularly strong as it is given by a mixture of point masses; we also show how this helps to make sense of various observed phenomena about ensembles.

**What this paper is and what it is not**  In this work we establish a simple yet precise and formal connection between deep ensembles and empirical Bayes, showing that ensembles can be interpreted as a particular way of carrying out Bayesian inference for neural networks. This result reveals that deep ensembles are a type of BNN and that these two paradigms are thus not fundamentally distinct. Although we hope that this new understanding of the connection between BNNs and ensembles will lead to enhanced UQ in the future, our goal here is only to establish this connection and to link it to existing work.

## 2 Background

Throughout this paper we will consider a dataset $\mathcal{D}$ along with a probabilistic model $p(\mathcal{D} \mid \theta)$ describing how the observed data is assumed to depend on some parameter $\theta \in \Theta$ of interest. Here $\theta$ are the parameters of a neural network, and $\Theta$ the set of possible parameter values. For example, in the classification setting with i.i.d. data, $\mathcal{D} = \{(x_i, y_i)\}_{i=1}^n$, where $x_i$ represents a feature vector which is treated as a covariate, $y_i$ is a categorical label, and $p(\mathcal{D} \mid \theta) = \prod_i p(y_i \mid \theta, x_i)$, where $p(y_i \mid \theta, x_i)$ is the probability assigned by the neural network classifier to $y_i$ when given $x_i$ as input. This is the most commonly studied setting for UQ and we will use it throughout our work. Nonetheless, we highlight that our main result applies to any setting with a well-defined likelihood $p(\mathcal{D} \mid \theta)$.[1] In the rest of this section we cover the preliminaries needed for our main result: BNNs, empirical Bayes, variational inference, and deep ensembles.

### 2.1 Bayesian Neural Networks

Most learning algorithms produce a single parameter estimate $\theta^*$, for example, maximum-likelihood achieves this by maximizing $\log p(\mathcal{D} \mid \theta)$ over $\theta \in \Theta$. For a query point $x_{n+1}$, the resulting predictive distribution $p(\cdot \mid \theta^*, x_{n+1})$ can be used not only for prediction,[2] but it also aims to encapsulate the *aleatoric uncertainty* (i.e. irreducible uncertainty) associated with predicting the label of $x_{n+1}$.

In contrast with most learning algorithms, Bayesian methods aim to obtain a distribution over $\theta$ which can then be leveraged for improved prediction and UQ. At a high level, BNNs operate by first specifying a prior distribution $\pi$ on $\Theta$ and then following the rules of Bayesian inference to obtain the corresponding posterior distribution $\pi(\cdot \mid \mathcal{D})$, given by

$$\pi(\theta \mid \mathcal{D}) \propto \pi(\theta)\, p(\mathcal{D} \mid \theta). \tag{1}$$

This posterior encapsulates *epistemic uncertainty*, i.e. uncertainty arising from a lack of perfect knowledge of the underlying true data-generating process. As such, variability in $p(\cdot \mid \theta, x_{n+1})$ over values of $\theta$ sampled from the posterior can be used to quantify epistemic uncertainty. The posterior can also be combined with the predictive distribution to obtain the posterior predictive,

$$p(\cdot \mid x_{n+1}) = \mathbb{E}_{\theta \sim \pi(\cdot \mid \mathcal{D})} \left[ p(\cdot \mid \theta, x_{n+1}) \right]; \tag{2}$$

---

[1]For example, $y_i$ could easily represent a regression target instead of a classification label, or in the setting of unsupervised i.i.d. generative modelling, $\mathcal{D} = \{x_i\}_{i=1}^n$ and $p(\mathcal{D} \mid \theta) = \prod_i p(x_i \mid \theta)$.

[2]For a function $f$, we use the notation $f(\cdot)$ to refer to the function $f$ itself, not $f$ evaluated at an input "·". Thus, for example, $p(\cdot \mid \theta^*, x_{n+1})$ refers to the actual probability mass function over labels, whereas $p(y \mid \theta^*, x_{n+1})$ corresponds to this probability mass function evaluated at the label $y$.

this is often called Bayesian or posterior averaging. By virtue of using both the predictive distribution and the posterior, the posterior predictive can be understood as aiming to capture *predictive uncertainty*, i.e. the overall uncertainty involved in the prediction of the label of $x_{n+1}$, comprising both aleatoric and epistemic uncertainties.

In practice computing the posterior and posterior predictive is highly intractable. This issue is typically circumvented by first obtaining $M$ samples $\theta_1, \ldots, \theta_M$ approximately distributed according to the posterior, which can then be used to produce a Monte Carlo estimate of any quantity of interest involving the posterior. For example, the posterior predictive can be estimated as

$$p(\cdot \mid x_{n+1}) \approx \frac{1}{M} \sum_{m=1}^{M} p(\cdot \mid \theta_m, x_{n+1}). \tag{3}$$

We will see how variational inference (VI; Wainwright & Jordan, 2008; Kingma & Welling, 2014; Rezende et al., 2014; Blei et al., 2017) can be used to obtain $\theta_1, \ldots, \theta_M$ in Subsection 2.3 and how this relates to ensembles in Section 3.

## 2.2 Empirical Bayes

Empirical Bayes is a class of procedures where the prior is learned from $\mathcal{D}$, rather than fixed beforehand in an attempt to express prior uncertainty about $\theta$. Empirical Bayes is a sensible approach towards setting the prior when fixing a prior through expert knowledge is challenging, as is the case in BNNs (Wenzel et al., 2020); it is also principled (Carlin & Louis, 2000) and can result in superefficient estimators (James & Stein, 1961; Efron & Morris, 1971; 1972). When the prior is learnable, we will consider a set of potential priors $\Pi \subset \Delta(\Theta)$, where $\Delta(\Theta)$ denotes the set of distributions on $\Theta$. In this work we will focus on a particular empirical Bayes procedure, namely maximum marginal likelihood, which maximizes $\log p_\pi(\mathcal{D})$ over $\pi \in \Pi$, where

$$p_\pi(\mathcal{D}) = \mathbb{E}_{\theta \sim \pi}[p(\mathcal{D} \mid \theta)] \tag{4}$$

is called the marginal likelihood. We will denote the optimal prior as $\pi^*$ and the corresponding posterior as $\pi^*(\cdot \mid \mathcal{D})$. Directly optimizing the marginal likelihood is intractable, and VI can also be used towards this goal. We will show in Section 3 how deep ensembles implicitly use VI to perform empirical Bayes.

## 2.3 Variational Inference

As previously mentioned, exact Bayesian inference and direct maximum marginal likelihood are both intractable; VI allows for simultaneously performing approximate Bayesian inference and marginal likelihood maximization. VI starts by introducing a family of distributions $\mathcal{Q} \subset \Delta(\Theta)$. The premise of VI is to simultaneously find $\pi^* \in \Pi$ through maximum marginal likelihood, along with an element $q^* \in \mathcal{Q}$ such that $q^*$, often called the variational posterior, best approximates the corresponding true posterior, i.e. $q^* \approx \pi^*(\cdot \mid \mathcal{D})$. Formally, this is achieved by introducing the evidence lower bound (ELBO),

$$\text{ELBO}(q, \pi) := \mathbb{E}_{\theta \sim q}\left[\log p(\mathcal{D} \mid \theta)\right] - \mathbb{KL}\left(q \,\|\, \pi\right) = \log p_\pi(\mathcal{D}) - \mathbb{KL}\left(q \,\|\, \pi(\cdot \mid \mathcal{D})\right), \tag{5}$$

and then obtaining $q^*$ and $\pi^*$ by maximizing the ELBO over $(q, \pi) \in \mathcal{Q} \times \Pi$. Note that the ELBO is defined through the first equality in Equation 5, which can be tractably maximized when $\mathcal{Q}$ is adequately chosen. The second equality in Equation 5 shows why the ELBO provides a sensible objective, even though neither $\log p_\pi(\mathcal{D})$ nor $\mathbb{KL}(q \,\|\, \pi(\cdot \mid \mathcal{D}))$ can be individually evaluated nor straightforwardly approximated: maximizing the ELBO indeed promotes the concurrent maximization of the marginal likelihood and the matching between the variational and true posteriors.

VI is not always used in the context of empirical Bayes; when the prior is fixed, the ELBO is only maximized over $q \in \mathcal{Q}$, which simply approximates the true posterior distribution. The family of distributions $\mathcal{Q}$ is typically chosen so that sampling from its elements is straightforward. Therefore, regardless of whether the prior is learned or not, once $q^*$ has been obtained, we can simply independently sample $\theta_1, \ldots, \theta_M \sim q^*$ to then approximate the posterior predictive as in Equation 3.

### 2.4 Deep Ensembles

Deep ensembles (Lakshminarayanan et al., 2017) are very simple: they train $M$ models through maximum-likelihood, resulting in parameter values $\theta_1^*, \ldots, \theta_M^*$, and average the models. All these models are trained through the same stochastic gradient-based procedure, for example Adam (Kingma & Ba, 2015), except optimization trajectories differ across models due to randomization. The predictive distribution implied by this procedure, which we denote as $p_{\text{ens}}(\,\cdot\mid x_{n+1})$, can then be written as

$$p_{\text{ens}}(\,\cdot\mid x_{n+1}) \coloneqq \frac{1}{M}\sum_{m=1}^{M} p(\,\cdot\mid \theta_m^*, x_{n+1}) = \mathbb{E}_{\theta\sim\pi_{\text{ens}}}\left[p(\,\cdot\mid \theta, x_{n+1})\right], \quad \text{where} \quad \pi_{\text{ens}}(\theta) \coloneqq \frac{1}{M}\sum_{m=1}^{M}\delta_{\theta_m^*}(\theta), \quad (6)$$

with $\delta_{\theta_m^*}$ representing a point mass at $\theta_m^*$.

The distribution $\pi_{\text{ens}}$ plays an analogous role to a Bayesian posterior in several ways: it also aims to encapsulate epistemic uncertainty, and Equation 6 is highly reminiscent of the posterior predictive in Equation 2 since both compute an expectation of $p(\,\cdot\mid \theta, x_{n+1})$ over $\theta$. However, although deep ensembles produce accurate and well-calibrated models, they are presumed to be less principled than BNNs since $\pi_{\text{ens}}$ is obtained in a seemingly *ad-hoc* way, whereas the posterior is derived through the principles of Bayesian inference. We also highlight that Equation 3 and Equation 6 result in very similar computations as they both average $p(\,\cdot\mid \theta, x_{n+1})$ over $M$ values of $\theta$, but the former does so to approximate an intractable expectation while the latter calculates its respective expectation exactly.

In summary, in spite of some similarities, BNNs and deep ensembles are often surmised to be fundamentally different. We will shortly challenge this view by showing how BNNs are linked to deep ensembles via empirical Bayes and VI.

## 3   How Deep Ensembles Perform Exact Bayesian Averaging via Empirical Bayes

In this section we will present our main result: that deep ensembles can be derived as implicitly performing maximum marginal likelihood through VI with an arbitrarily flexible prior and flexible enough variational posterior. We begin by stating a simple observation which we will use during our discussion.

**Observation 1.** *Let $\Theta_c$ denote the c-level set of the likelihood for $c > 0$, i.e. $\Theta_c \coloneqq \{\theta \in \Theta : p(\mathcal{D}\mid\theta) = c\}$. Assume $\Theta_c$ is non-empty and let $\pi_c \in \Delta(\Theta)$ be a probability distribution placing all its mass on $\Theta_c$, i.e. $\mathbb{P}_{\theta\sim\pi_c}(\theta \in \Theta_c) = 1$. Then, $\pi_c = \pi_c(\,\cdot\mid\mathcal{D})$.*

*Proof.* Recall that the support of a posterior distribution must be contained in the support of its corresponding prior, and thus the proportionality in Equation 1 holds over the support of the prior, i.e. $\pi_c(\theta\mid\mathcal{D}) \propto \pi_c(\theta)p(\mathcal{D}\mid\theta)$ is valid over the support of $\pi_c$. Since by assumption the support of $\pi_c$ is contained in $\Theta_c$, it follows that[3]

$$\pi_c(\theta\mid\mathcal{D}) \propto \pi_c(\theta)p(\mathcal{D}\mid\theta) = \pi_c(\theta)c \propto \pi_c(\theta). \tag{7}$$

Since two distributions can be proportional to each other only if they are equal, the equation above implies that $\pi_c = \pi_c(\,\cdot\mid\mathcal{D})$. □

Note that Observation 1 can be informally restated as saying that if the likelihood does not depend on $\theta$ over the support of the prior, then the posterior matches the prior. We will soon see that the prior implicitly learned by deep ensembles is actually supported on a set where the likelihood does not depend on $\theta$.

We now formally state the assumptions behind our main result.

---

[3]Note that in Equation 7 we are treating the prior and the posterior as densities even though these distributions need not admit a density with respect to some standard base measure (e.g. the Lebesgue or counting measures). There is nonetheless no error in Equation 7 when both the prior and posterior are understood as densities with respect to the same (potentially not standard) base measure on $\Delta(\Theta)$.

**Assumption 1.** *The likelihood function $p(\mathcal{D} \mid \cdot) : \Theta \to [0, \infty)$ achieves its maximum, i.e. there exists at least one parameter value $\theta^* \in \Theta$ such that $p(\mathcal{D} \mid \theta) \leq p(\mathcal{D} \mid \theta^*)$ for every $\theta \in \Theta$. We denote the set of all such likelihood maximizers as $\Theta^*$. Aditionally, we assume the likelihood is strictly positive under any of these maximizers, i.e. $p(\mathcal{D} \mid \theta^*) > 0$.*

**Assumption 2.** *The learnable prior is arbitrarily flexible in the sense that $\Pi = \Delta(\Theta)$.*

**Assumption 3.** *The variational posterior is flexible enough in the sense that if $\pi^* \in \Pi$ maximizes the marginal likelihood $p_\pi(\mathcal{D})$ from Equation 4, i.e. $p_\pi(\mathcal{D}) \leq p_{\pi^*}(\mathcal{D})$ for every $\pi \in \Pi$, then $\pi^*(\cdot \mid \mathcal{D}) \in \mathcal{Q}$.*

Assumption 1 is sensible, and for example holds whenever the model is not pathological in the sense that at least one parameter value assigns positive likelihood to the observed data, the likelihood is continuous in $\theta$, and $\Theta$ is compact (which is implicitly enforced whenever weight clipping is used). Note that we are not assuming that the likelihood maximizer is unique; indeed, neural networks being overparameterized suggests $\Theta^*$ should have many (potentially infinitely many) elements. For example, various works have empirically found the existence of contiguous regions in the parameter space of neural networks corresponding to optimal (or near optimal) loss values (Draxler et al., 2018; Garipov et al., 2018; Benton et al., 2021; Şimşek et al., 2021). These works highlight that, in contrast to settings in traditional statistics where the likelihood has a unique maximizer, $\Theta^*$ is a complex set which is given by (or at least contains) a low-dimensional submanifold of $\Theta$. We highlight as well that the complexity in $\Theta^*$ goes beyond statistical non-identifiability, i.e. different parameters $\theta^* \in \Theta^*$ can result in different predictive distributions $p(\cdot \mid \theta^*, x_{n+1})$; we will see shortly how this complexity allows ensembles to quantify epistemic uncertainty.

Note also that Assumption 2 and Assumption 3 need not hold when a learnable prior or variational posterior are instantiated as neural networks; rather, these two assumptions are needed to characterize the optima of the ELBO under an idealized highly flexible VI procedure, which we will shortly link to deep ensembles. We carry out this characterization in our main result below.

**Proposition 1.** *Let $(q^*, \pi^*) \in \mathcal{Q} \times \Pi$. Then, under Assumption 1, Assumption 2, and Assumption 3, $(q^*, \pi^*)$ maximizes the ELBO in Equation 5, i.e. $ELBO(q, \pi) \leq ELBO(q^*, \pi^*)$ for every $(q, \pi) \in \mathcal{Q} \times \Pi$, if and only if the following two properties hold:*

*(A) $q^*$ places all of its mass on $\Theta^*$, i.e. $\mathbb{P}_{\theta \sim q^*}(\theta \in \Theta^*) = 1$.*

*(B) The prior and variational posterior match, i.e. $\pi^* = q^*$.*

*Proof.* We begin by proving that if $(A)$ and $(B)$ hold, then $(q^*, \pi^*)$ maximizes the ELBO. Let $(q, \pi) \in \mathcal{Q} \times \Pi$ and assume $(A)$ and $(B)$ hold. We have:

$$\text{ELBO}(q, \pi) = \mathbb{E}_{\theta \sim q}[\log p(\mathcal{D} \mid \theta)] - \mathbb{KL}(q \,\|\, \pi) \leq \mathbb{E}_{\theta \sim q}[\log p(\mathcal{D} \mid \theta)] \leq \mathbb{E}_{\theta \sim q^*}[\log p(\mathcal{D} \mid \theta)] \qquad (8)$$

$$= \mathbb{E}_{\theta \sim q^*}[\log p(\mathcal{D} \mid \theta)] - \mathbb{KL}(q^* \,\|\, \pi^*) = \text{ELBO}(q^*, \pi^*), \qquad (9)$$

where the last inequality in Equation 8 follows from $(A)$ and the first equality in Equation 9 follows from $(B)$. Thus, $(q^*, \pi^*)$ maximizes the ELBO. The equations above also show that the maximal value achieved by the ELBO is $\mathbb{E}_{\theta \sim q^*}[\log p(\mathcal{D} \mid \theta)]$, which by Assumption 1 is neither $-\infty$ nor $\infty$.

We now assume that $(q^*, \pi^*)$ maximizes the ELBO and will prove that $(A)$ and $(B)$ hold. We proceed by contradiction and assume that $(B)$ does not hold, meaning that $\mathbb{KL}(q^* \,\|\, \pi^*) > 0$. Since the maximal value of the ELBO is finite it follows that $-\infty < \mathbb{E}_{\theta \sim q^*}[\log p(\mathcal{D} \mid \theta)] < \infty$, and we then have that

$$\text{ELBO}(q^*, \pi^*) = \mathbb{E}_{\theta \sim q^*}[\log p(\mathcal{D} \mid \theta)] - \mathbb{KL}(q^* \,\|\, \pi^*) < \mathbb{E}_{\theta \sim q^*}[\log p(\mathcal{D} \mid \theta)] - \mathbb{KL}(q^* \,\|\, q^*) \qquad (10)$$

$$= \text{ELBO}(q^*, q^*). \qquad (11)$$

By Assumption 2, $q^* \in \Pi$, so the above equations show that $(q^*, \pi^*)$ does not maximize the ELBO. This is a contradiction and thus $(B)$ must hold. It remains to prove that $(A)$ also holds, which we now do.

Let $\hat{\pi} \in \Delta(\Theta)$ be a probability distribution placing all of its mass on $\Theta^* \subset \Theta$ and let $\pi \in \Pi$, so that by Assumption 1, $p_\pi(\mathcal{D}) = \mathbb{E}_{\theta \sim \pi}[p(\mathcal{D} \mid \theta)] \leq \mathbb{E}_{\theta \sim \hat{\pi}}[p(\mathcal{D} \mid \theta)] = p_{\hat{\pi}}(\mathcal{D})$ with $0 < p_{\hat{\pi}}(\mathcal{D}) < \infty$. It follows that $\hat{\pi}$ maximizes the marginal likelihood, and by Assumption 3, $\hat{\pi}(\cdot \mid \mathcal{D}) \in \mathcal{Q}$. We now also proceed by contradiction and assume $(A)$ does not hold, so that $p_{\pi^*}(\mathcal{D}) < p_{\hat{\pi}}(\mathcal{D})$. We then have two cases:

1. $\mathbb{KL}\left(q^* \parallel \pi^*(\cdot \mid \mathcal{D})\right) < \infty$, which implies that

$$\mathrm{ELBO}(q^*, \pi^*) = \log p_{\pi^*}(\mathcal{D}) - \mathbb{KL}\left(q^* \parallel \pi^*(\cdot \mid \mathcal{D})\right) < \log p_{\hat{\pi}}(\mathcal{D}) - \mathbb{KL}\left(q^* \parallel \pi^*(\cdot \mid \mathcal{D})\right) \tag{12}$$

$$\leq \log p_{\hat{\pi}}(\mathcal{D}) = \log p_{\hat{\pi}}(\mathcal{D}) - \mathbb{KL}\left(\hat{\pi}(\cdot \mid \mathcal{D}) \parallel \hat{\pi}(\cdot \mid \mathcal{D})\right) = \mathrm{ELBO}\left(\hat{\pi}(\cdot \mid \mathcal{D}), \hat{\pi}\right). \tag{13}$$

2. $\mathbb{KL}\left(q^* \parallel \pi^*(\cdot \mid \mathcal{D})\right) = \infty$, in which case the strict inequality in Equation 12 does not hold anymore as both sides would be $-\infty$. Nonetheless, Assumption 1 ensures that in this case we still have that

$$\mathrm{ELBO}(q^*, \pi^*) = -\infty < \log p_{\hat{\pi}}(\mathcal{D}) = \mathrm{ELBO}\left(\hat{\pi}(\cdot \mid \mathcal{D}), \hat{\pi}\right). \tag{14}$$

In short, $\mathrm{ELBO}(q^*, \pi^*) < \mathrm{ELBO}(\hat{\pi}(\cdot \mid \mathcal{D}), \hat{\pi})$ holds in both cases. By Assumption 2, $\hat{\pi} \in \Pi$, and so it follows that $(q^*, \pi^*)$ does not maximize the ELBO. This is again a contradiction, and thus $(A)$ holds. $\square$

Proposition 1 characterizes optimal priors $\pi^*$ according to maximum marginal likelihood under an arbitrarily flexible prior model $\Pi$: these priors must place all their mass on $\Theta^*$, the set of maximum-likelihood estimators. Crucially, this is exactly how deep ensembles set $\pi_{\mathrm{ens}}$ in Equation 6: $\pi_{\mathrm{ens}}$ clearly assigns probability one to $\Theta^*$ since each ensemble member is trained to convergence, so that $\pi_{\mathrm{ens}}$ is indeed an optimal prior as prescribed by Proposition 1. Furthermore, if we write $c^* = \max_{\theta \in \Theta} p(\mathcal{D} \mid \theta)$, which is well-defined thanks to Assumption 1, then $\Theta_{c^*} = \Theta^*$ and by Observation 1, it follows that $\pi_{\mathrm{ens}}$ is not only equal to an empirical Bayes prior, but it is also equal to the corresponding posterior, i.e. $\pi_{\mathrm{ens}} = \pi_{\mathrm{ens}}(\cdot \mid \mathcal{D})$.[4] In turn, $\pi_{\mathrm{ens}}$ is an actual posterior distribution, rather than simply playing an analogous yet *ad-hoc* role to one as commonly thought. It also follows from this reasoning that $p_{\mathrm{ens}}(\cdot \mid x_{n+1})$ is an actual posterior predictive distribution, an thus deep ensembles perform *exact* Bayesian averaging. In summary, deep ensembles are equivalent to a particular way of implicitly performing empirical Bayes through maximum marginal likelihood; this justifies ensembles and formally links them to BNNs.

The insights arising from Proposition 1 go beyond merely interpreting deep ensembles through empirical Bayes. Whenever $\Theta^*$ has more than a single element, there are infinitely many optimal priors $\pi^*$ assigning all their probability mass to $\Theta^*$, and deep ensembles simply find one such prior. These priors are optimal in the sense of maximizing the marginal likelihood, but Proposition 1 does not provide a criterion to determine which of all these priors will result in optimal UQ. Note that despite this optimality, priors placing all their mass on $\Theta^*$ need not properly quantify epistemic uncertainty. For example, the prior $\delta_{\theta^*}$, where $\theta^* \in \Theta^*$, is optimal from an empirical Bayes perspective yet quantifies no uncertainty over $\theta$. Thanks to the complexity of $\Theta^*$, ensembles can be simultaneously empirical-Bayes-optimal while also inducing non-trival uncertainty over $\theta$. In this sense, it is rather remarkable that ensembles succeed at UQ given how strong their prior is; indeed, $\pi_{\mathrm{ens}}$ essentially corresponds to the simplest empirical Bayes prior satisfying these optimality and non-triviality conditions. Intuitively, selecting for the most diverse empirical Bayes priors might improve UQ and thus lead to empirical improvements. It could also be argued that all these priors, despite being optimal from the perspective of maximizing marginal likelihood, are too strong as they trivialize Bayesian inference since they are equal to their corresponding posteriors. It might thus also be the case that following an empirical Bayes procedure where $\Pi$ is restricted in such a way that the corresponding ELBO-maximizing $\pi^*$ concentrates most of its mass around $\Theta^*$ – as opposed to all of its mass exactly on $\Theta^*$ – would also empirically improve UQ. Nonetheless, we highlight that instantiating a method to tractably implement any of these potential improvements is not trivial and falls outside of the scope of our work.

## 4 Relationship to Existing Work

**Deep ensembles** Despite approaching a decade of existence, deep ensembles (Lakshminarayanan et al., 2017) remain a gold standard for UQ with neural networks, with most follow-up work focusing on improving

---

[4]It is well-known that maximizing the ELBO results in the variational posterior matching the true posterior when $\mathcal{Q}$ is flexible enough; this remains true in the setting of Proposition 1 where the empirical Bayes prior matches the variational posterior since these two distributions are both also equal to the true posterior. Note also that in practice each ensemble member might achieve slightly different loss values, or equivalently, likelihoods $p(\mathcal{D} \mid \theta_m^*)$. In this case, although the posterior $\pi_{\mathrm{ens}}(\cdot \mid \mathcal{D})$ will not exactly match the prior $\pi_{\mathrm{ens}}$, it will still be very similar to it as long as all ensemble members obtain similar likelihoods (this happens because the posterior has the same support as the prior and is obtained by weighting each ensemble member proportionally to its likelihood). Thus, our derived understanding of ensembles still applies when not all members achieve exactly the same loss.

computational efficiency rather than performance (Huang et al., 2017; Garipov et al., 2018; Wen et al., 2020; Havasi et al., 2021). As previously mentioned, ensembles are often explicitly described as non-Bayesian; our work disproves this conventional understanding and provides intuitions that will hopefully lead to future performance improvements. Wu & Williamson (2024) proposed one of the rare methods whose goal is to outperform deep ensembles at UQ. Interestingly, despite not connecting ensembles and empirical Bayes, their motivation is precisely that $\pi_{\text{ens}}$ being a mixture of point masses is overly strong. We thus see the good empirical results of Wu & Williamson (2024) as evidence supporting the intuitions developed from our newfound understanding of deep ensembles. We also highlight the works of Abe et al. (2022) and Abe et al. (2024), who challenge the common view that ensembles succeed at UQ due to the predictive diversity of their averaged models. These works conclude that although ensembles improve upon the accuracy and UQ of their base models, a single larger model matching the ensembles' accuracy will also match its performance at UQ. In other words, ensembling is a good way to increase accuracy and more accurate models tend to be better at UQ; it is this increase in accuracy, rather than predictive diversity, which drives the empirical improvements in UQ obtained by ensembling. These findings are once again consistent with the intuition we have developed: using a prior such as $\pi_{\text{ens}}$ which concentrates all its mass on $\Theta^*$ is likely overly strong and may well be suboptimal in terms of the predictive diversity that it induces.

**BNNs** Bayesian methods to express uncertainty over the weights of neural networks take various forms. Some works use VI by instantiating a restrictive variational family $\mathcal{Q}$ so as to make direct maximization of the ELBO tractable (Graves, 2011; Blundell et al., 2015; Louizos & Welling, 2016; 2017; Wu et al., 2019; Osawa et al., 2019). Our view of deep ensembles deviates from these BNNs in that the variational posterior implicitly used by ensembles is much more flexible. Other approaches forgo the use of VI altogether, for example by using Markov chain Monte Carlo to approximately sample from the posterior (Welling & Teh, 2011; Chen et al., 2014; Zhang et al., 2020). Using a second-order Taylor expansion of the log posterior results in a Gaussian approximation known as Laplace approximation; this idea has also been leveraged in the context of Bayesian deep learning (Ritter et al., 2018; Kristiadi et al., 2020; Daxberger et al., 2021). All the aforementioned BNNs use simplistic priors such as centred Gaussians with identity or diagonal covariance; this is a key difference between common BNN methods and our empirical Bayes view of ensembles, which allows for much richer priors. An interesting consequence of using such flexible priors is that ensembles are the *only* BNNs which perform exact posterior averaging – all the other BNNs mentioned above rely on approximations in one way or another.

**Monte Carlo dropout** Dropout (Srivastava et al., 2014) was first proposed as a regularization technique where weights are randomly set to zero with a certain probability $\omega$ during training. Gal & Ghahramani (2016) showed that using dropout at test time can be loosely interpreted as sampling from a variational posterior corresponding to a simple fixed prior. The resulting method, Monte Carlo (MC) dropout, is thus often considered Bayesian, although the approximation it makes to the variational posterior is so crude that this interpretation has been called into question (Le Folgoc et al., 2021). Here we make the observation that our work provides an alternative interpretation of MC dropout as performing empirical Bayes: the distribution $q^*$ used by MC dropout at test time (corresponding to the trained network with dropout turned on) is learned by attempting to maximize $\mathbb{E}_{\theta \sim q}[\log p(\mathcal{D} \mid \theta)]$ over $q \in \mathcal{Q}$, where $\mathcal{Q}$ is the set of possible distributions over parameters implied by dropout.[5] In turn, $q^*$ should be expected to place most of its mass around $\Theta^*$, suggesting that using $q^*$ as a prior should result in a posterior which is also very similar to $q^*$ – this justifies using $q^*$ for Bayesian averaging.

**Empirical Bayes in modern machine learning** Empirical Bayes has uses in machine learning beyond quantifying uncertainty in neural network weights. For example, in the context of variational autoencoders (Kingma & Welling, 2014; Rezende et al., 2014) learning flexible priors is a widespread practice which improves performance (Sønderby et al., 2016; Chen et al., 2017; Tomczak & Welling, 2018; Dai & Wipf, 2019; Pang et al., 2020; Vahdat & Kautz, 2020; Child, 2021; Vahdat et al., 2021; Loaiza-Ganem et al., 2022; 2024). Unfortunately, porting these methods over to BNNs is challenging, since the very high dimensionality of $\Theta$ renders them intractable. In Gaussian processes, learning prior hyperparameters by maximizing the

---

[5]Formally, this $\mathcal{Q}$ is given by $\mathcal{Q} = \{q \in \Delta(\Theta) : q(\theta') = \prod_j ((1 - \omega)\delta_{\theta_j}(\theta'_j) + \omega\delta_0(\theta'_j))$ for some $\theta \in \Theta\}$, where the product runs over all the coordinates of the parameter.

marginal likelihood is equally prevalent (Rasmussen & Williams, 2005; Titsias, 2009; Wilson et al., 2016; Gardner et al., 2018), and empirical Bayes is also useful in the context of denoising generative models (Saremi & Hyvärinen, 2019). Wenzel et al. (2020) performed a large-scale study finding, among other things, that the use of simplistic priors is a potential cause for the underperformance of Bayesian averaging in BNNs. Several of the BNN methods mentioned above use empirical Bayes to learn the variance of their Gaussian prior; doing so commonly results in empirical improvements over using fixed isotropic Gaussians as priors. These improvements are both unsurprising in light of the benefits brought forth by empirical Bayes, and consistent with the results of Wenzel et al. (2020) since these learned Gaussian priors are slightly less simplistic than isotropic ones. The difference between how deep ensembles and these other BNNs apply empirical Bayes lies once again in the flexibility of the learnable prior. Importantly, our interpretation of deep ensembles offers a simple potential explanation for their empirical success over other BNNs: when learning the prior, it is likely better to be overly flexible rather than overly restrictive. We nonetheless highlight that the reason this flexibility helps is unclear; it could simply be that a more complex $\Pi$ allows to better express prior uncertainty and thus results in improved UQ, but it could also be the case that the improvements come from ensembles circumventing the need for approximate Bayesian averaging. We believe that further exploring these ideas is a compelling avenue for future work.

**On the relationship between deep ensembles and BNNs** Connections between BNNs and ensembles have been studied before, albeit not from the perspective of empirical Bayes. Wild et al. (2024) use Wasserstein gradient flows (Ambrosio et al., 2008) to study an idealized version of the optimization dynamics of VI and deep ensembles; they conclude that ensembles are not Bayesian unless ensemble members are trained jointly and an explicit term encouraging diversity is added to their training objective (D'Angelo & Fortuin, 2021). Note that there is no contradiction with our results here since the prior is fixed in the work of Wild et al. (2024) whereas it is learnable in our Bayesian interpretation of deep ensembles. Pearce et al. (2020) also proposed a training modification to ensembles wherein models are regularized towards parameter values sampled from the prior, which they argue renders ensembles Bayesian. Closer to our work is that of Wilson & Izmailov (2020), who claim that deep ensembles perform approximate Bayesian averaging. They argue that for common choices of prior (e.g. Gaussian) and likelihood used in BNNs, the resulting posterior $\pi(\,\cdot\mid\mathcal{D})$ will naturally concentrate most of its mass around $\Theta^*$, so that the maximum-likelihood estimates $\theta_m^*$ used by ensembles can be viewed as approximate samples from $\pi(\,\cdot\mid\mathcal{D})$; when this holds, the averaging done by ensembles in Equation 6 to obtain $p_{\text{ens}}(\,\cdot\mid x_{n+1})$ can indeed be understood as an approximation of the Monte Carlo estimate in Equation 3 (which is itself an approximation of the true posterior predictive $p(\,\cdot\mid x_{n+1})$ in Equation 2). The connection between deep ensembles and BNNs established by Wilson & Izmailov (2020) provides sensible intuitions, and empirical evidence suggests that $p_{\text{ens}}(\,\cdot\mid x_{n+1})$ can indeed be close to $p(\,\cdot\mid x_{n+1})$ (Izmailov et al., 2021). Nonetheless we highlight that in contrast to ours, this connection is only approximate, not rigoruous, and assumes the prior is fixed beforehand.

## 5   Conclusions

In this paper we introduced a simple and fresh perspective for reasoning about deep ensembles and their relationship to BNNs. Our work serves as a position paper about understanding ensembles, and in contrast with previous work which has viewed ensembles as either non-Bayesian or merely approximating posterior averaging, we showed that through the lens of empirical Bayes, ensembles are the only BNN which performs exact posterior averaging. Our developed understanding of ensembles is consistent with and expands upon existing literature. Lastly, our work suggests that explicitly learning flexible priors within BNNs is likely to produce empirical improvements in UQ. Although doing so in a tractable way is not trivial due to the very large number of parameters in modern neural networks, we remain hopeful that this issue will be circumvented in future research.

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
