# OpenReview forum: "Deep Ensembles Secretly Perform Empirical Bayes"
_TMLR — Rejected by TMLR_

### Review · Reviewer_cgNr · 2025-02-17

**Summary Of Contributions:**

The paper derives an intriguing formal connection between ensembles and Bayesian Deep Learning methods.
Specifically, they prove that independently optimizing the members of an ensemble using maximum likelihood estimation produces both the empirical Bayes prior and posterior.

**Audience:**

Yes

**Broader Impact Concerns:**

I do not foresee any concerns with this paper.

**Claims And Evidence:**

Yes

**Requested Changes:**

I will try to summarize my requested changes below:

* The initial claim that ensembles lack a formal connection to BNNs comes off a bit too strong in my opinion; while the authors (to the best of my knowledge) are the first to prove this connection explicitly, other works like the later cited Wilson & Izmailov (2020) and Wild et al. (2024) have already argued in this direction.
* I would appreciate a slightly longer discussion about the practical implications of the proven results, e.g. which practices that modelers use might break any of the posed assumptions.
* The work by Pearce et al. (http://proceedings.mlr.press/v108/pearce20a/pearce20a.pdf) seems relevant, as it discussed and also formally argues how minor modifications to ensembles result in approximartely Bayesian ensembling.
* There are a couple of commas missing throughout the text:
    * In the abstract: "In other words, deep ensembles"
    * At the end of page 3: "once q* has been obtained, we can simply"
    * Towards the end of page 5: "By Assumption 2, q*..."
    * First paragraph of page 6: "... and by Observation 1, it follows that..."

**Strengths And Weaknesses:**

The motivation as well as the proofs are laid out in a very readable and understandable manner.
No empirical results are given, although I personally belief that this is not strictly necessary for theoretical results, especially in a journal submission. The paper tries to not overstay its welcome and is kept pretty compact, nevertheless I felt that the practical implications of this result were not entirely clear to me after the reading.

---

> ### Author Response · Authors · 2025-03-05
> **Rebuttal**
>
> Thank you for your feedback and for the time you spent reviewing our paper. We are glad you found our paper “intriguing” and “very readable and understandable”. We have updated our manuscript in a way that we believe addresses the points you raised (changes are highlighted in blue). We updated our claim about the connection between BNNs and ensembles, we mentioned the work of Pearce et al. in the related work section, and added the missing commas.
>
> As for the discussion about practical implications: we believe that our work can lead to improved uncertainty quantification in BNNs since it suggests that explicitly learning highly flexible priors through empirical Bayes is a good idea. More specifically, since ensembles are implicitly doing this, it is also likely to help in BNNs; additionally, the prior used by ensembles is so strong that, intuitively, using slightly weaker priors in BNNs might produce improvements over ensembles. This being said, learning a flexible prior through empirical Bayes is not straightforward in BNNs. For example, in the setting of variational autoencoders, flexible priors are often learned through empirical Bayes by being instantiated as normalizing flows or by having a hierarchical structure. However, these approaches are completely intractable in the setting of BNNs since the dimension of the problem is the number of network parameters. In short, although we believe that using flexible learnable priors within BNNs is possible, it is not straightforward. Thus, in our view, proposing a tractable method to instantiate learnable flexible priors within BNNs would be its own contribution; this is why we decided to not include experiments in the paper. We have also expanded upon this point in our manuscript updates.
>
> Please let us know if you have any lingering concerns.

---

### Review · Reviewer_D2Xb · 2025-02-24

**Summary Of Contributions:**

This paper shows that deep ensembles can be interpreted as empirically Bayesian. Each maximum likelihood training of a model can be thought of as learning a prior distribution with a point mass around the maximum likelihood estimate. Then, the posterior distribution is trivially the same as the prior distribution due to support based arguments.

**Audience:**

Yes

**Claims And Evidence:**

No

**Requested Changes:**

Section 2.4: cite the original Lakshminarayanan paper.

Discussion of what the level set of likelihood maximizers for NNs looks like would be nice. This is alluded to in the references to Garipov et al, ’18. Several other works explore the construction of this set in other ways – Draxler et al, ’18, Benton et al, ’21, Simsek et al, ‘21

Both Zhang et al, ’20 and Izmailov et al, ’21 use multiple independent chains in their approximate posterior inference, which ends up looking quite like the deep ensemble solution but expanded to level sets of the likelihood.

“clear empirical improvements…” I’m not sure this is actually always the case, see for example Appendices E and F of Wilson & Izmailov, ’20.

The much earlier work of Pearce et al, ’20 is empirically Bayesian as well as using ensembles to justify the mixture of point mass posteriors, so discussion with that work would be pretty interesting as it ends up being a kind of variational approximation.

Discussion with Wu & Williamson, ‘24’s connection of ensembles to the Bayesian bootstrap, another type of approximate posterior would also be quite useful in connecting this paper to the wider literature.

References:

Draxler et al, “Essentially No Barriers in Neural Network Energy Landscape”, ICML, 2018.

Benton et al, : Loss Surface Simplexes for Mode Connecting Volumes and Fast Ensembling, ICML, 2021.

Simsek et al, “Geometry of the Loss Landscape in Overparameterized Neural Networks: Symmetries and Invariances”, ICML, 2021.

Pearce et al, : Uncertainty in Neural Networks: Approximately Bayesian Ensembling, AISTATS, 2020.

**Strengths And Weaknesses:**

Strengths:

This is a clean argument that does justify the case for using deep ensembles in a Bayesian method in practice.

The argument itself seems mathematically correct, conditional on the assumptions.

Weaknesses:

The broad argument of the assumptions is that we train the neural network to reach the maximizer of the log-likelihood and then the variational posterior (and prior) is the set of all maximizers of the log-likelihood. However, this is not true in practice for cross-entropy loss as we never reach -infinity loss on the training set (the maximal loss value); for regression (maybe with GPs or something), this might be possible under perfect interpolation for NNs. For convex problems, the prior and posterior is just a single point mass.

Observation 1: uniform on the level set, this is a prior that has probability mass zero in continuous space. This seems very reliant on hitting precisely a log-likelihood of \log c, which each ensemble value may or may not reach. What happens if this is relaxed to be continuous with \theta_c = {\theta: p(\D | \theta) >= c}?

Ultimately, the argument is tautologically true – the posterior for deep ensembles is “exact” because the prior we set to be whatever we learned from optimization. It’s a nice “justification” but definitely unsettling in practice – by this logic we can claim anything we want is “empirically” Bayesian. The more usual argument is that the posterior is a mixture of point masses centered around the independent models, but this is obviously only correct at a handwavy level as Wild et al, ’24 point out.

---

> ### Author Response · Authors · 2025-03-05
> **Rebuttal (1/2)**
>
> We thank you for your feedback and for the time you spent reviewing our submission. We are glad you found that we present a “clean argument that does justify the case for using deep ensembles in a Bayesian method in practice”. Below we address the issues you raised one by one:
>
> - You claimed that ` the argument is tautologically true [...] by this logic we can claim anything we want is “empirically” Bayesian` and that `a model can be thought of as learning a prior distribution with a point mass around the maximum likelihood estimate. Then, the posterior distribution is trivially the same as the prior distribution due to support based arguments`
>
> We believe that these claims arise from a slight misunderstanding of our argument. It is of course indeed the case that, if a prior is given by a *single* point mass, then the corresponding posterior will be trivially equal to the prior. In this sense, you are also correct that anything is empirically Bayesian, if by “anything” we understand a single point estimate represented by a corresponding point mass. This is of course not an interesting setting though, since the whole point of Bayesian inference is to quantify (epistemic) uncertainty over $\theta$ (i.e. network parameters), and assuming a prior which is given by a *single* point mass is assuming no such uncertainty in the value of $\theta$.
>
> However our result goes beyond point masses. For example, the papers you shared (that we are now referencing in our updated manuscript) essentially show that $\Theta^*$ (i.e. set of maximum-likelihood-optimal parameters) is given by a low-dimensional submanifold of parameter space. Our work highlights that *any* prior whose support is contained within this submanifold – *even if this prior does not assign positive probability to any single parameter value* (i.e. does not contain any point masses) – will result in a posterior which matches the specified prior.
>
> Our argument about ensembles is that ensembles can be understood as one particular way of instantiating a prior whose support is contained in $\Theta^*$ – *this is the property which allows us to interpret them as empirically Bayesian, not the fact that each member could be understood as its own point mass prior*. Importantly, the prior in ensembles is a mixture of point masses rather than a single point mass, and although this prior is still simple, it does entail uncertainty in the values of $\theta$ and thus does not trivialize like the single point mass case.
>
> You also mention that “the more usual argument is that the posterior is a mixture of point masses centered around the independent models, but this is obviously only correct at a handwavy level as Wild et al, ’24 point out” – we note that this mixture of point masses is also our view of the posterior in ensembles, but we show that this view is only handwavy when the prior is fixed (e.g. Gaussian), and that it can be formally derived through empirical Bayes since $\pi_{\text{ens}}$ satisfies the optimality criteria established in Proposition 1 (i.e. it assigns probability $1$ to $\Theta^*$).
>
> We have now updated our manuscript to better convey these points.
>
> - You mentioned that one of our assumptions is that ` we train the neural network to reach the maximizer of the log-likelihood [...] this is not true in practice for cross-entropy loss as we never reach -infinity loss on the training set`
>
> We believe that the loss you are referring to is the function space loss (which should achieve a value of $0$ at optimality), whereas we are talking about parameter space. For a fixed architecture, the best achievable loss over parameters $\theta$ need not correspond to the best loss achievable with arbitrarily flexible classifiers (i.e. function space). As we mention in the paper, an example where our assumption holds is the case where the likelihood is continuous and $\Theta$ is compact. Additionally, as long as ensembles members achieve the same likelihood (or very similar, as argued in the following point), then they belong to the same level set and thus can still be understood as an empirical Bayes procedure wherein the data is used to find a prior (even if this prior is not optimal from a marginal likelihood point of view). In short, we believe this assumption reasonably describes common practical settings.

---

> > ### Author Response · Authors · 2025-03-05
> > **Rebuttal (2/2)**
> >
> > - You asked, about Observation 1,  `this seems very reliant on hitting precisely a log-likelihood of \log c [...] what happens if this is relaxed to be continuous with \theta_c = {\theta: p(\D | \theta) >= c}?`
> >
> > Thank you for raising this, you are indeed correct that in practice all ensemble members need not achieve the exact same loss. Although we do not believe that Observation 1 can be extended to other non-level-set subsets of $\Theta$ in general, we can still answer what the consequences of slightly different losses are for ensembles. More precisely, let’s still denote the ensemble parameters as $\theta_m^*$, even if they do not achieve the exact same likelihood. Assuming a uniform prior over the $M$ ensemble members, $\pi(\theta) = \tfrac{1}{M}\sum_{m=1}^M \delta_{\theta_m^*}(\theta)$, following Bayes’ Theorem results in a posterior which is supported on $\{\theta^*_1, \dots, \theta^*_M\}$ given by
> >
> > $\pi(\theta_m^* \mid \mathcal{D}) = \dfrac{p(\theta_m^* \mid \mathcal{D})}{\sum_{m'=1}^M p(\theta_{m'}^* \mid \mathcal{D})}$,
> >
> > i.e. each ensemble member $\theta^*_m$ is assigned posterior probability proportional to its likelihood. When all these likelihoods are identical, the posterior assigns the same mass to each member and thus matches the prior. When these likelihoods are slightly different, although the posterior will not formally match the prior, the corresponding probabilities assigned to each ensemble member will still be close to $1/M$. Although it is indeed the case that in practice each member will achieve slightly different loss, the losses will be very similar, and the entailed posterior very close to matching the prior. Although one could reweight ensemble predictions by their likelihoods to recover this posterior, this is likely to have no practical effect due to how similar the likelihood values are. In short, even when ensemble members achieve slightly different likelihoods, we believe that the understanding that we derived remains a useful way to think about ensembles.
> >
> > - Among your requested changes, you mentioned `Both Zhang et al, ’20 and Izmailov et al, ’21 use multiple independent chains in their approximate posterior inference, which ends up looking quite like the deep ensemble solution but expanded to level sets of the likelihood.` and `Discussion with Wu & Williamson, ‘24’s connection of ensembles to the Bayesian bootstrap, another type of approximate posterior would also be quite useful in connecting this paper to the wider literature.`
> >
> > We will happily expand upon our discussions of these related works, but could you please clarify the discussions that you had in mind? Although the works of Zhang et al (2020) and Izmailov et al (2021) indeed resemble ensembling in their use of several MCMC chains (the use of several chains is common practice in the MCMC literature), we do not immediately see how this connection goes beyond the “usual”, informal, connection between ensembling and Bayesian averaging, nor how our interpretation of ensembling as empirical Bayes connects to Bayesian inference with fixed priors when several MCMC chains are used. Similarly, could you please elaborate on the connection between Bayesian bootstrap in the work of Wu & Williamson (2024) that you would like us to explore further? We also do not immediately see a concrete connection with empirical Bayes here.
> >
> >
> > We have updated our manuscript to include our replies to the points you raised, as well as the changes, citations, and discussions you requested (changes are highlighted in blue). Please let us know if you have any lingering concerns.

---

> > > ### Comment · Reviewer_D2Xb · 2025-03-09
> > > **More clarifications**
> > >
> > > > we believe that the understanding that we derived remains a useful way to think about ensembles.
> > >
> > > Yes, I believe that your understanding is useful but most useful when considering some sort of set that is actually continuous (e.g. the prior corresponding to all likelihood values \geq c). In that case, even prior sampling is probably a good way to sample the posterior as you correctly pointed out.
> > >
> > > > Zhang et al, '20, Izmailov et al, '21
> > >
> > > Both of these methods roughly speaking train several independent models (ensembles) and then "sample" from the posterior near the end of training, after a low loss / high likelihood value has been reached. We can interpret this as uniform sampling from the set of all likelihood values \geq c for example. As they're covering multiple models, this is actually empirical validation of the extension of your argument I was asking about.
> > >
> > > > Wu & Williamson, '24
> > >
> > > The bayesian bootstrap is a reasonably well justified way of getting ensembles that are explicitly approximately Bayesian inference. The method proposed in that paper targets a slightly weaker posterior, the martingale posterior.

---

> > > > ### Author Response · Authors · 2025-03-11
> > > > **Second response (1/2)**
> > > >
> > > > Thank you very much for your reply and for engaging in discussion with us, we appreciate the added clarification about your concerns. We reply to your points below, please let us know if any of your concerns still remain.
> > > >
> > > > - re: the mixtures of point masses used by ensembles are `in some very real sense still a point mass in the much higher dimensional space because the functional forms of these models are completely identical. Weight space symmetries for example produce this type of behavior (Brea et al, '19).`
> > > >
> > > > While this point may hold true in a regime with infinite data, we do not believe it holds in practice for two reasons. First, weight symmetries imply that *some* optimal parameters are functionally equivalent, not that all of them are. There is no mathematical basis to believe that there is a unique function achieving an optimal loss for a given finite dataset, and we should actually expect infinitely many such functions to exist (e.g. infinitely many distinct polynomials can achieve optimal loss in regression, and similarly infinitely many distinct functions will achieve perfect classification accuracy as long as they correctly classify the finite training data).
> > > >
> > > > Second, if it was indeed the case that ensembles correspond to a point mass on function space, then they would not do a good job at quantifying uncertainty in practice. More specifically, it is well documented in the literature (see the many references in the paper) that the variability in weight space of ensemble members actually translates into variability in the corresponding predictions. This variability in the predictions would not happen if ensemble members were functionally equivalent, and it is precisely this variability that allows ensembles to succeed at quantifying uncertainty.
> > > >
> > > > Lastly, we wonder if your underlying concern when you asked about extending our result to different level sets is that the variability in the predictions of ensemble members does not stem from the existence of various optimal solutions in function space, but rather from the optimum not being exactly achieved during optimization, thus causing differences in the trained models? If so, we highlight that some of the variability in ensembles’ predictions arising from optima not being perfectly achieved is compatible with the point we are making about non-uniqueness causing this variability. That being said, as discussed above, we do believe that most of this variability does arise as a consequence of the non-uniqueness of optimal solutions.
> > > >
> > > > - You also mentioned that ` I believe that your understanding is useful but most useful when considering some sort of set that is actually continuous`.
> > > >
> > > > We believe we might be in agreement here, please see the discussion in the last paragraph of section 3. More specifically, note that the level sets as we defined them (i.e. likelihood being exactly equal to $c$) can also be continuous. The works you brought up about $\Theta^*$ being a submanifold of parameter space back this up. If your point is that placing a mixture of point masses on $\Theta^*$ is too simple a way of placing probability $1$ on $\Theta^* and that some continuous density over this submanifold might achieve better results than ensembles, we actually agree: we are claiming that empirical Bayes justifies and allows us to interpret ensembles as Bayesian, but that it is not necessarily optimal as a way to quantify uncertainty.

---

> > > > > ### Author Response · Authors · 2025-03-11
> > > > > **Second response (2/2)**
> > > > >
> > > > > - You mentioned that `And note that continuous \theta is typically how NNs are trained, which does violate some of your assumptions (" the likelihood is continuous and \theta is compact. ") - of course, we can trivially clamp \theta but this is worth pointing out.`
> > > > >
> > > > > We would like to clarify that we are not claiming anywhere that optimization of neural networks is not carried over continuous parameters. We agree that compactness does not hold without some sort of clamping though, but actually it is not uncommon to apply clamping to weights or gradients during optimization (note that clamping gradients while also having a pre-defined maximal number of gradient steps that can be taken implies a bound on the explorable space of parameters, essentially implying compactness). We also highlight that we see this point as somewhat tangential. The assumption of perfect optimization is of course idealized (as are all theoretical assumptions), and we make it to be able to derive a mathematical understanding of ensembles through empirical Bayes. In practice, we expect small deviations from this assumption to only result in slight deviations from its corresponding implications. In other words, we can still understand ensembles as closely approximating their idealized version where all members achieve the exact same loss.
> > > > >
> > > > > - Thank you for clarifying the discussion points about the works of Zhang et al. (2020), Izmailov et al. (2021), and Wu & Williamson (2024) that you wanted us to discuss further. We will include these points in the final version of our paper.

---

> > ### Comment · Reviewer_D2Xb · 2025-03-09
> > **Thanks for the clarifications**
> >
> > Thank you for the comments and clarifications, a few notes here:
> >
> > >We believe that the loss you are referring to is the function space loss (which should achieve a value of at optimality), whereas we are talking about parameter space.
> >
> > For continuous \theta and a large enough NN, these should roughly correspond. And note that continuous \theta is typically how NNs are trained, which does violate some of your assumptions (" the likelihood is continuous and \theta is compact. ") - of course, we can trivially clamp \theta but this is worth pointing out.
> >
> > > [the prior is .. ] a low-dimensional submanifold of parameter space
> >
> > Indeed, this is the argument that you're making. However, this is in some very real sense still a point mass in the much higher dimensional space because the functional forms of these models are completely identical. Weight space symmetries for example produce this type of behavior (Brea et al, '19). So, it's not super clear to me whether or not these are actually different than a special definition of a point mass. By comparison, the level set argument does induce different point masses and an actually continuous prior in the full dimensional set of all parameters.
> >
> >
> > References:
> > Brea et al, 2019 - https://arxiv.org/abs/1907.02911

---

### Review · Reviewer_nAnP · 2025-02-28

**Summary Of Contributions:**

In the present work the authors explore the association between two state-of-the-art methods employed for uncertainty quantification (UQ) in deep learning, namely Bayesian Neural Networks (BNNs) and deep ensembles. The authors theoretically illustrate that deep ensembles perform exact Bayesian averaging through implementing an empirical Bayes procedure in which the prior is learned from the data.
Such finding is useful towards understanding that deep ensembles are not ad-hoc but are rather supported by theoretical justification related to Bayesian theory

**Audience:**

Yes

**Claims And Evidence:**

Yes

**Requested Changes:**

The authors are welcomed to add an experiments section in which they illustrate their theoretical findings

**Strengths And Weaknesses:**

Strengths :

-- theoretical justification of deep ensembles;
-- insights on why deep ensembles illustrate such strong empirical performance;
-- analysis of deep ensembles learned prior: it is shown that it is given by a mixture of point masses

Weaknesses

-- although the theoretical results do help towards understanding the performance of deep ensembles , the paper lacks of an experiments section that supports/explains the findings

---

> ### Author Response · Authors · 2025-03-05
> **Rebuttal**
>
> Thank you very much for your feedback and for the time you spent reviewing our paper. Your one criticism was the lack of experiments in our paper. Although we agree that experiments showing that our findings can be leveraged to improve upon ensembles and/or BNNs would strengthen our work, we highlight that doing so is not trivial. More specifically, our work suggests that using explicitly learnable, highly flexible priors in BNNs is likely to improve their performance. Conceptually this is easy, for example this is commonly done in variational autoencoders using normalizing flow priors or hierarchical priors. However, using these approaches in the setting of BNNs is completely intractable due to the very high dimensionality of the parameter space of neural networks. We believe this tractability issue can be circumvented, but doing so requires additional research which is separate from the understanding of ensembles that we aimed to derive in our paper. We thus believe that methodology and experiments based on our insights would constitute their own contributions which are better left for future work. We have modified our manuscript to more explicitly mention this point (changes are highlighted in blue).
>
> Lastly, we believe that, even without experiments, our paper satisfies TMLR’s acceptance criteria: (1) we think our derived connection between ensembles and BNNs is of clear interest to a subset of TMLR’s audience, and (2) we provide evidence for our claims in the form of precise and formal mathematical statements.

---

> > ### Comment · Reviewer_nAnP · 2025-03-05
> >
> > Dear authors,
> >
> > Thanks for the clarifications
> > I am, of course, aware of TMLR acceptance criteria; my point was that, in my opinion,  adding an experiments section would strengthen the paper (and its contribution), and  would also help towards providing empirical findings and insights
> > Nevertheless, I do understand your argument on the difficulty of this task
> >
> > Aa orthogonal, additional comment is that there's abuse of notation in equation (2) ; addressing it would be appreciated

---

> > > ### Author Response · Authors · 2025-03-05
> > > **Reply**
> > >
> > > Thank you very much for your engagement!
> > >
> > > As for the abuse of notation in equation 2, are you referring to the fact that we use "$\cdot$" for both labels $y$ and parameters $\theta$ as input? If so, note that we understand the notation $f(\cdot)$ as equivalent to the function $f$ itself, not to $f$ evaluated at the input "$\cdot$". While this distinction is trivial when there is a single variable, it becomes useful when a function can have more than a single input, as is the case in the paper. With this interpretation of the notation, we do not believe there is any abuse of notation. Please let us know if we misunderstood your point though; we will happily update the notation if we misunderstood, or add a footnote clarifying the notation otherwise.

---

> > > > ### Comment · Reviewer_nAnP · 2025-03-05
> > > >
> > > > In this case I would suggest that maybe it would be useful to define/explain the notation as well, as it could be confusing for some readers

---

> > > > > ### Author Response · Authors · 2025-03-05
> > > > > **Updated manuscript to clarify notation**
> > > > >
> > > > > Thanks again for helping improve the clarity of our paper, we have added a footnote (footnote 2) clarifying our notation.

---

### Decision · Action_Editor_m7VT · 2025-04-28

**Recommendation:** Reject

**Comment:**

This paper addresses a hot and controversial discussion of whether deep ensembles can be considered as some form of approximate Bayesian inference.  Deep ensembles have been shown to perform exceptionally well (at least compared to various more sophisticated methods) at uncertainty quantification in deep learning.  As they involve averaging over multiple parameter settings, they are reminiscent of Bayesian model averaging.  The discussion in the subfield of uncertainty quantification is whether these can be considered to be approximate Bayesian inference, and thus inherit the benefits of Bayesian methods (compositionality, theoretical guarantees, etc.), or heuristic / ad-hoc.   This paper presents an interpretation, somewhat of a theoretical note, that ensembles perform an empirical Bayes procedure where the parameters of the prior are learned from the data as a mixture of point masses.  The posterior then is somewhat trivially matched to the prior.

The work is interesting and well written and brings a novel perspective to this highly discussed topic.  Two of the reviewers are "leaning accept" but one reviewer recommended reject.  This reviewer found the the argument to be unconvincing and thus found the claims of the paper to not be justified by evidence.  I find this reviewers argument compelling.  Performing empirical Bayes to learn the prior over many parameters seems rather un-Bayesian.  It seems like the authors try to pin down a theory that fits deep ensembles within the Bayesian paradigm.  However, the conclusion doesn't really seem to follow from the theory and one might argue that it points the conclusion in the other direction - i.e. if ensembles were Bayesian then the prior would have to be a mixture of point masses optimized over the data, and the posterior would match this prior.  The argument is a bit too much of a stretch given that there are no experiments to empirically back it up.  Agreed that it seems challenging to do so in large scale models.  However, using relatively small models, one could compare the ensembles to a gold-standard HMC implementation.

Unfortunately, the decision is to reject the paper since it seems not quite ready for publication.  The argument presented is interesting, novel and potentially exciting, but it doesn't seem quite ready to upend the discussion as claimed by the authors.  I would encourage the authors to continue developing the argument and see if they can provide some empirical evidence as well, and then resubmit.

**Audience:**

Whether deep ensembles are Bayesian is a controversial topic within a fairly large subfield of deep learning.  Strong arguments in either direction would certainly be of interest to the community.  However, misleading arguments then would serve to only further confuse the discussion.

**Claims And Evidence:**

Two reviewers found that the claims were justified by evidence while one reviewer did not think so.  The reviewer who disagreed found that the authors' arguments weren't convincing.  This paper presents a theoretical argument that addresses a popular and controversial question in Bayesian deep learning, i.e. whether deep ensembles can be interpreted as being some form of approximate Bayesian inference.  In this case, the authors present a theory that the deep ensembles are performing some sort of empirical Bayes on a lower bound of the marginal likelihood.  The reviewer argues that the presented theory is somewhat tautologically true and doesn't imply that deep ensembles are Bayesian.  There are no experiments to validate the theory.

**Resubmission Of Major Revision:**

The authors may consider submitting a major revision at a later time.